# Perceptions and Practices of General Practitioners towards Oral Cancer and Emerging Risk Factors among Indian Immigrants in Australia: A Qualitative Study

**DOI:** 10.3390/ijerph182111111

**Published:** 2021-10-22

**Authors:** Nidhi Saraswat, Rona Pillay, Neeta Prabhu, Bronwyn Everett, Ajesh George

**Affiliations:** 1Centre for Oral Health Outcomes and Research Translation (COHORT), Ingham Institute for Applied Medical Research, School of Nursing and Midwifery, South Western Sydney Local Health District, Western Sydney University, Liverpool, NSW 1871, Australia; B.Everett@westernsydney.edu.au (B.E.); a.george@westernsydney.edu.au (A.G.); 2School of Nursing and Midwifery, Western Sydney University, Parramatta, NSW 2116, Australia; Rona.Pillay@westernsydney.edu.au; 3Paediatric Dentistry, Faculty of Medicine and Health, School of Dentistry, The University of Sydney, Camperdown, NSW 2006, Australia; neeta.prabhu@sydney.edu.au; 4Paediatric Dentistry, Westmead Centre for Oral Health, Sydney, NSW 2145, Australia; 5Faculty of Medicine and Health, School of Dentistry, The University of Sydney, Camperdown, NSW 2006, Australia

**Keywords:** oral cancer, knowledge, awareness, beliefs, perceptions, clinical practices, general practitioners, Indian immigrants, Australia

## Abstract

Background: In Australia, Indian immigrants are one of the fastest-growing communities. Since oral cancer is widespread in India, the indulgence of Indians in customs of areca (betel) nut use in Australia may be linked to the recent rise in oral cancer cases. Since GPs (general practitioners) are primary healthcare providers, it is pivotal to ensure the oral cancer awareness of GPs. This study aimed to explore oral cancer risk-related knowledge, beliefs, and clinical practices of GPs in Australia. Methods: Fourteen semi-structured interviews were conducted with GPs practicing across New South Wales and Victoria. Purposive and snowball sampling were used for recruitment. Data were analysed through a directed content analysis approach. Results: All GPs were knowledgeable of major oral cancer causative factors including tobacco and alcohol, but some had limited understanding about the risks associated with areca nut preparations. Positive attitudes were evident, with all participants acknowledging the importance of oral cancer risk assessment. Most GPs recalled not performing oral cancer routine check-ups. Conclusion: GPs presented good oral cancer knowledge except for emerging risk factors such as areca nut use. Varied beliefs and inconsistent clinical practices relating to oral cancer screening is concerning. Accessible oral cancer training around emerging risk factors may benefit GPs.

## 1. Introduction

Oral health is a fundamental component of general health and wellbeing. Poor oral health significantly impacts a person’s quality of life [1] and poses a major health burden leading to pain, discomfort, infection, and even death [2]. Despite being largely preventable, oral diseases affect approximately 3.5 billion people worldwide and remain mostly neglected within general health policy [1]. Furthermore, several systemic diseases such as AIDS (Acquired Immuno-Deficiency Syndrome) are also present with early signs of lesions in the oral cavity [3]. These factors highlight the importance of routine oral examination as a vital step for the early detection of various diseases [4]. However, the widespread perception that oral diseases and lesions are limited to the scope of dental practice [5] has caused the marginalisation of oral health from mainstream developments in healthcare systems [1]. Although dentists have a definitive and unique role in diagnosing oral cancer [6], the high cost of dental treatments deters patients from visiting dentists on a regular basis [7]. In addition, in most regions of the world including developed countries, access to affordable oral healthcare is limited and aggravated by poor awareness about oral diseases [8]. To address this, general practitioners (GPs) and many other non-dental health professionals (for example—physicians, physician assistants, nurses, nurse practitioners) may have potentially crucial roles to play in the prevention and management of oral diseases [8]. Importantly, GPs are often the primary contact for patients, and routine examination of the oral cavity by GPs could considerably reduce the morbidity resulting from serious oral diseases [4].

A particularly serious oral disease is oral cancer, which comprises 1–2% of all cancers that may arise in the body [9]. Oral cancer is among the top 15 most common cancers worldwide [1,10], and the global incidence of cancers of the lip and oral cavity had been estimated around 377,713 new cases for the year 2020 [11]. Oral cancers are generally chronic, and they are usually asymptomatic until an advanced stage [12]. They are generally associated with high mortality rates and expensive medical treatment [5]. When compared to other major types of cancers, oral cancer has one of the lowest five-year survival rates (50% or less in reported cases) [13]. However, survival rates of 70% to 90% can be achieved if this disease is diagnosed and treated at early stage [13]. Furthermore, oral cancer patients with early-stage disease often require less intervention [12]. This is also important because high-risk patients developing oral cancer, for example, the older, heavy tobacco and alcohol users are often irregular dental attendees and more likely to attend hospitals for medical needs [14]. Therefore, early detection in primary healthcare settings is a key to reduce diagnostic delay [15], and GPs can play a crucial role in the early identification of oral cancer through a mouth/oral cavity examination with basic equipment [16].

Oral cancer has a high prevalence in South Asia and is on the rise in other countries [10]. The leading risk factors for oral cancer are tobacco use, alcohol consumption, and areca (betel) nut chewing along with poor dietary habits [1,2]. In the past few decades, a rapid rise in incidence of HPV (Human Papilloma Virus)-associated oral cavity cancers has also been noticed in many developed countries [17]. It is a well-known fact that chronic alcohol and tobacco smoking can lead to delayed wound healing, which in turn increases the susceptibility to various infections, including oral cancer lesions [18]. On the other hand, South Asian countries including India, Pakistan, and Bangladesh have a long history of chewing areca nut preparations [10], which is also a socially accepted culturally-embedded custom in the Indian subcontinent [19]. Areca (betel) nut has carcinogenic potential, and areca nut chewing has been found to be significantly associated with poor prognosis in patients diagnosed with oral cancer [20]. India is considered the epicentre of oral cancer disease with the highest number of cases reported each year [21]. Given that India is the largest source of immigrants [22], the elevated risk associated with the above practices is suspected to be carried by Indians to other parts of the world through immigration [19,23,24]. It is also noteworthy that a sharp increase in oral cancer incidence has been reported in the past decade in several countries that are popular for new immigrants including Denmark, France, Germany, Scotland and to some extent in Australia, New Zealand, the United Kingdom, and the United States [3].

In Australia, oral cancer represents less than 3% of all cancer cases, yet it continues to pose a significant disease burden due to the poor survival rates [25]. Mortality associated with oral cancer has remained stable over the past three decades despite a decrease in incidence over the last two decades [25]. More than 4000 new cases of head and neck cancers, including lip, are diagnosed every year in Australia, and almost 600 of these cancers are oral cavity cancers [26,27]. Among other factors responsible, this rise in numbers has also been linked to tobacco chewing and cultural practices of areca nut chewing within the Indian immigrant communities in Australia [28,29]. This is concerning, since India is the top source country of immigrants with 25,698 places (approximately 18.3%) in Australian immigration [30], and Indians are one of the fastest-growing communities forming about 2.6% of the total population in Australia [31]. With the increasing oral cancer incidence rates [26] and changing immigration patterns [32], it becomes even more important to ensure oral cancer awareness of GPs to assist in early identification and public education. This is particularly important in Australia, as recent studies indicate that only half of all oral health professionals are performing oral cancer screening with their patients [33].

However, not much is known about GPs’ oral cancer-related knowledge and clinical practices in Australia, especially around emerging oral cancer risk practices among new immigrants. Exploring this information is important as a recent review regarding oral cancer-related practices of GPs in developed countries revealed limited oral cancer knowledge around newly emerging causative factors and a lack of related clinical training for routine oral cancer screening and/or counselling [34].

Thus, this study aimed to explore the oral cancer-related knowledge, beliefs, and clinical practices of GPs in Australia. This study was part of a broader mixed-method study that investigated the knowledge, attitudes, and practices of Indian immigrants towards oral cancer, the findings of which are published elsewhere [35].

The following research questions guided this study:What are the oral cancer-related knowledge, beliefs and clinical practices of GPs in Australia?What are the perceived barriers for GPs regarding oral cancer risk assessment and counselling of patients?What are GPs’ recommendations to promote oral cancer awareness particularly among high-risk populations such as Indians in Australia?

## 2. Materials and Methods

### 2.1. Design

This exploratory study used a qualitative research design involving interviews with GPs. Exploratory qualitative research has been identified as a suitable method to study areas (specifically within healthcare practice) that have previously received little or no attention [36]. Semi-structured interviews were chosen to provide scope to use prompts to draw out additional information and to clarify responses. This method of data collection was appropriate, as this study intended to explore oral cancer-related knowledge, beliefs, and clinical practices of GPs in Australia. The Consolidated Criteria for Reporting Qualitative Studies (COREQ): 32-item checklist [37] was used to report the qualitative aspect of this research (see Appendix A).

### 2.2. Sampling and Recruitment

All the GPs practicing in the suburbs of New South Wales (NSW) and Victoria (VIC), Australia were eligible to participate. A combination of purposive and snowball sampling was used to enhance participation. Since the larger study also investigated the oral cancer awareness needs of Indian immigrants, suburbs of Sydney and Melbourne were the preferred recruitment sites, as they are densely populated and represented the required Indian population [38]. No exclusion criteria were set based on age, gender, country of origin or number of years of clinical practice in Australia. The email contacts of medical practices in these areas were retrieved through information publicly available on the internet. An invitation to participate with contact details was emailed to all practices, and a follow-up reminder email was sent after one week in case of no response. Those GPs who expressed an interest in participating were provided with a participant information sheet outlining the purpose of the study, and interviews were scheduled at a convenient time. All attempts were made to ensure diversity in the sample in terms of gender, age, country of birth, and educational training. To become a general practitioner in Australia, one must either complete an undergraduate double degree or a bachelor’s degree followed by a postgraduate degree to become registered with the Medical Board of Australia [39]. Overseas trained GPs have to go through a comprehensive assessment process before registration to ensure their qualifications (undergraduate or postgraduate) are comparable to the Australian standards [40]. Once registered, GPs have the option of undertaking additional specialist training in general practice or other specialities through a fellowship [39].

### 2.3. Data Collection

A literature review [34] and interviews with Indian immigrants (conducted as part of a larger study) [35] informed the development of an interview guide, which was further refined with a multidisciplinary team involved in this research (see Appendix A—Interview focus areas and guiding questions). Recruitment and data collection continued until data saturation, where no new information emerged from the interviews (*n* = 14). Previous studies have recommended that a minimum sample size of at least 12 is sufficient to reach data saturation in qualitative studies [41,42]. Furthermore, a recent systematic review of interview-based qualitative studies found eight to ten participants was appropriate in studies that were exploratory in nature [43]. The principal researcher (NS, female, Ph.D. candidate) was experienced in qualitative methods and conducted all 14 interviews between August and November 2019 with one interview face-to-face and the remaining 13 interviews conducted over the telephone. Written or verbal consent was obtained.

Due to time constraints and the busy schedules of GPs, the interviews varied in length ranging from 17 min (shortest interview) to 43 min (longest interview). The interviews were audio-recorded and then professionally transcribed. The transcripts were not returned for member checking due to the time constraints of participants. Demographics of the participants including age, gender, highest level of qualification, and years of clinical practice in Australia were recorded. Participants were also given an opportunity at the end of the interview to add any further comments not addressed in the interview.

### 2.4. Data Analysis

Transcripts were checked by the principal researcher (NS) against the original audio recordings for accuracy and then imported into qualitative data management software (QSR NVivo 12 (QSR International, Melbourne, VIC, Australia)). The data were analysed using directed qualitative content analysis (QCA). A formative categorisation matrix based on the themes identified from an integrative review of oral cancer-associated knowledge, attitudes, and practices of South Asian immigrants [24] guided the initial coding. All authors (Ph.D. candidate (NS), two practicing/academic dentists (AG and NP), and two nurse academics (RP and BE)) undertook the initial coding by reading two transcripts each. Over the course of two meetings, consensus was reached regarding a coding structure, and this was used by NS to code the remaining transcripts. After the initial analysis was completed, the first author (NS) went back through the coded excerpts and identified subcategories, which were then presented to two other researchers in the team (AG and RP).

### 2.5. Ethical Considerations

The study received ethics approval from the Human Research Ethics Committee of Western Sydney University (H13203). All GPs were given a gift voucher (AU$50) as reimbursement for their time. The audio recordings and transcripts were stored on a password protected computer as per institutional and ethics committee requirements.

### 2.6. Rigor

Rigor was maintained at every stage of data collection and analysis to enrich the study. Several methodological strategies were employed to address trustworthiness—the criteria for vigorous qualitative research (credibility, transferability, dependability and confirmability). An individual interview format was selected to facilitate participants’ disclosure of relevant information without any confidentiality concerns. Debriefings were organised with another researcher (AG) to discuss the completeness of data. For accuracy, a professional transcription company was employed for verbatim transcriptions of the interview audio recordings. Participants were de-identified throughout transcription to ensure their anonymity. To enhance credibility, all researchers in the team did individual coding, and then, consensus was achieved with subsequent team meetings. Codes and subcategories were robustly discussed and confirmed with two other researchers (namely, (RP) and (AG)). To ensure transferability, detailed information is provided about the study settings, participants, and data collection. Direct quotes by the participants have been used to support the findings. The implementation of these methodological strategies has ensured the rigor and trustworthiness of this research.

## 3. Definition of Terms

The terms ‘knowledge’, ‘beliefs’, and ‘clinical practices’ have been used widely in this paper. For the purposes of this paper, the definition of ‘knowledge’ refers to one’s awareness, level of understanding, and information in relation to oral cancer [34]. The term ‘beliefs’ has been used to depict the perceptions, views, and attitudes associated with oral cancer [34]. The ‘clinical practices’ of the participants (GPs in this context) relates to clinical actions encompassing oral cancer identification, counselling, and prevention [34]. ‘Risk products’ has been used as a term to represent commonly consumed tobacco and/or areca (betel) nut preparations [44]. In this paper, the term ‘Immigrant’ refers to a person who moves into a country other than that of his/her nationality [45].

## 4. Results

Fourteen GPs practicing across Sydney South (*n*  =  6), West (*n* = 3), North-West (*n* = 2), and East (*n* =  1), in New South Wales and Melbourne North (*n* = 1) and South-West (*n* =  1) in Victoria, Australia participated in this qualitative study. GPs originated from different countries with a handful from the Indian subcontinent (*n* = 5). Of the 14 GPs, nine were male participants and seven were within the age range of 35–54 years (range 25–64 years). More than half had obtained their basic qualifications from overseas (*n* = 9) and had undertaken specialist general practice training in Australia (*n* = 9). Their clinical practice experience in Australia ranged from 3 to 35 years (mean 10.7 years) (see Table 1 for demographics, given below). These demographics were fairly similar to the trends observed among GPs in Australia. Recent workforce data indicate there are more male GPs than female GPs in Australia (60% vs. 40%), more than half (53%) are within the age groups of 35–54 years, and the majority (51–80% depending on speciality training) have their basic qualifications from overseas [46]. In addition, according to the Australian Bureau of Statistics, there has been a marked increase in the number of GPs and specialists from South Asia [47], and they represent the second largest group of GPs after Australian-born general practitioners [48].

Three main categories and nine subcategories were identified from data analysis (see Table 2 for categories and subcategories, given below).

### 4.1. Oral Cancer-Related Knowledge

Overall, GPs demonstrated good knowledge and awareness about oral cancer risk.

#### 4.1.1. Recognition of Symptoms and Risk Factors

All participants were well informed about the main clinical signs—for example non-healing ulcer or an oral swelling—and knew the clinical symptoms such as bleeding or bad mouth odour commonly observed in oral cancer suspected cases. Major risk factors responsible for oral cancer such as tobacco and alcohol were also reported by every participant. However, varying levels of knowledge were apparent regarding the association of areca (betel) nut use and oral cancer. Three GPs were sceptical about the carcinogenicity of areca nut, while two had never heard of these products in Australia. As reflected from the following statements:


*I don’t think betel nut on its own is sufficient [to cause oral cancer]. You need all the other things in there that contributes to that…*
(GP-8)


*I haven’t heard of betel quid, specifically. I do know—as I said I know about smokeless tobacco… which, yeah, is a big risk factor…*
(GP-5)

More than half of the participants (*n* = 9) had prior training and clinical experience from overseas, particularly South Asia (*n* = 7) and were therefore more aware of areca nut usage as an emerging oral cancer risk factor as supported in the quote below:

*Because I am from an Indian background…. In India, it was more common because I guess people used to chew paan and gutka. So basically, that was like a betel nut product to be more precise which were the major risk factors of oral cancer*…(GP-2)

Moreover, a larger proportion of GPs (*n* = 9) also knew that some population groups such as South Asians and Indians were at high-risk for oral cancer owing to cultural practices:

*Middle East is also high risk. Indians, I think Indian background is high risk. Chinese because of the smoking probably, they’re also high risk*.(GP-6)

#### 4.1.2. Availability of Risk Products

Nearly all (*n* = 12) remarked that tobacco preparations and alcohol were readily available in Australia. In contrast, only GPs from Indian (*n* = 5) and Pakistani (*n* = 3) backgrounds were familiar with the availability of areca nut preparations at Asian/Indian grocery stores in Australia. For instance:

*I know the paans [betel quid] are available in a lot of the restaurants and these products are available in Indian grocery stores*.(GP-14)

The majority of GPs (*n* = 10) commented that areca nut preparations were more easily accessed in suburbs densely populated with Indian immigrants:


*I know that they’re [areca nut/betel quid] available in certain very limited areas where the ethnic population is predominately Indian...*
(GP-1)

There was some confusion though among interviewees (*n* = 6) over the legality of areca (betel) nut use in Australia:

*I think it’s legal. I don’t think that’s a—I don’t think it’s illegal in Australia*.(GP-8)

#### 4.1.3. Oral Cancer Training

All participants (*n* = 14) admitted not receiving any type of formal oral cancer-related training in Australia.

Nonetheless, eight GPs had some knowledge on oral cancer identification from their undergraduate course and clinical internship training overseas particularly from countries such as India and Pakistan where such habits are prevalent as noted in the following quote:

*In Australia no…In India yes. I used to do my internship in a government hospital and we did have a campaign for the doctors [about] how to identify and what to look for*.(GP-2)

Furthermore, several participants (*n* = 11) highlighted the lack of any specific resource related to oral cancer.


*I mean if someone has it [resource], there’s things we download, but I mean they’re not available unless I actually search for it...*
(GP-9)

### 4.2. Oral Cancer-Related Beliefs

Overall, positive beliefs and attitudes of GPs were recorded from the interviews.

#### 4.2.1. Views towards Oral Cancer Scenario in Australia

There were diverse views about the current oral cancer situation in Australia. Considering the low incidence and prevalence of oral cancer in Australia, six GPs were not sure if oral cancer could be a serious health issue in the near future:

*So, I guess cancer of the oral cavity in general in Australia is pretty low in terms of prevalence and incidents…even among the lower socioeconomic groups who are generally higher risk of smoking, and poorer oral hygiene, the numbers are still very small…I don’t think [it is] an emerging problem*.(GP-8)

In contrast, eight GPs were of the opinion that increasing immigration and continuation of habitual practices in Australia might lead to increased oral cancer risks in the future. As a few GP explained:

*Well, there might be [increased oral cancer risk] because Australia is having a lot of immigrants. With that, they bring likely new type of diets, and habits, and stuff. So, maybe it might increase the overall cancer risk*.(GP-14)

*But the way the migration is there, say 10, 15 years down the track, with the number increasing, it’s going to be more easy visibility of these paans and chewing tobacco which is sort of still available. Yes, I predict that [increased oral cancer risk] they will be*.(GP-13)

#### 4.2.2. Perceived Role in Oral Cancer Prevention

All participants believed that GPs have a crucial role in oral cancer prevention and early diagnosis as they are primary healthcare providers and generally the first to notice any suspicious oral lesions:

*I cannot stress enough the role of the GP, to be honest, as the GP is always the first point of contact. Because of the universal access to healthcare in Australia through Medicare, most of the patients, more or less, end up with a GP. Even if they know this particular matter is not related to the GP, but they know that the GP can direct them to the right person*.(GP-1)

*The role of the GP is important because I think you need the biopsy or the referral to a specialist because sometimes the patient will come in and then they will, obviously they don’t think it’s a throat cancer…I think without seeing a GP it’s very hard to diagnose it*.(GP-9)

Majority of the interviewees (*n* = 11) perceived GPs to play a preventive role in oral cancer care:

*So preventative care is good with GPs in case they find that even if they don’t have ulcers, but they are chewing tobacco, or they can be told about the risk factors and supported to quit*. (GP-11)

Many (*n* = 9) were of the belief that all patients including Indian immigrants would be open to receive oral cancer risk assessment or referrals and preventive education. In contrast, two GPs suggested that Indians may be reluctant in accepting patient counselling:

*I think—I find that generally, the Indian population, especially the migrant ones, they don’t—they’re not comfortable in [oral cancer prevention counselling]—they’ll only come when something is really bad. They won’t—in terms of prevention, they’re not very good with coming in for preventative stuff, they’re more—if it gets really bad and the symptoms don’t go away, they might come, they might be in a later stage*.(GP-9)

Another aspect highlighted by participants (*n* = 6) was the preference of some migrants to consult a GP from a similar cultural background, which could be an additional barrier for preventative strategies in this area:

*I feel that’s true, a lot of them, they do like to see—they say if you are an Asian you also want to see an Asian doctor, you know what I’m saying… I feel that it’s—they’re more open to that… Because there is cultural ethnicity factor*.(GP-4)

#### 4.2.3. Barriers to Oral Cancer Prevention and Management

Some participants (*n* = 8) emphasised a gap in the existing knowledge for oral cancer prevention and diagnosis with many practitioners referring patients to other specialties for a definitive diagnosis and opinion:

*From the GP’s point of view, I would say a knowledge gap; a lack of being able to do much except for refer. Probably a feeling that—I mean, the mouth is the area of either dentist or specialist in terms of our comfort to biopsy and comfort to manage oral changes and dental changes, and some uncertainty about where to send them I think*.(GP-10)

Limited time (*n* = 10) was cited as a major hurdle in oral cancer risk assessment and counselling:

*I think when it comes down to barrier, there’s no barrier, it’s just that we focus, as I said earlier as well, we focus more on the smoking part, but we always forget about it—I do try my best to do it, but the barrier is the time. If you’re running short of time and the waiting room is full of patients, so that’s one of the barriers*.(GP-13)

A small number of GPs (*n* = 3) highlighted the uncertainty while addressing oral concerns during check-ups owing to dentists having more expertise in this area as well as the high cost of dental treatments.

*The main barrier I have come across is sometimes there is a bit of overlap between dental and oral health issues. We recommend sometimes people do see the dentist and—because the dental assessment is quite often needed to make the better diagnosis. It could be an oral cancer issue. Sometimes people can’t afford seeing a dentist. That is quite a good barrier, I think*…(GP-14)

Other obstacles identified were lack of awareness (*n* = 3) and financial constraints (*n* = 2) among patients. One GP addressed this in the context of Indian immigrants as:

*You can call it ignorant behaviour. Yeah, that’s other way but that’s how I see is like they don’t see—they don’t see through actually. So, they’re not familiar with the health system…because most of the Indian patients I have to counsel them, they think why they are charging us the money*…(GP-3)

Various suggestions made by participants to address barriers in oral cancer prevention included oral cancer training endorsed by professional organisations such as the Royal Australian College of General Practitioners (RACGP), online Continuing Professional Development (CPD) activity (*n* = 9), educational pamphlets for clinical practice (*n* = 7), and raising awareness through social media and other advertisements (*n* = 10). Additionally, one GP even recommended incorporating oral cancer-training as part of the undergraduate medical curriculum:

*…So, that needs to be incorporated in the undergrad training. In the fellowship training, personally, which I did over here, they’re mostly focused on—they encourage GPs to identify population subgroups and then to know particular problems for them…but I’ve never received CPD activity for cancer. So, it’s kind of lack of effort on both hands*.(GP-1)

However, three participants were not in favour of additional oral cancer-related training for GPs:


*Because we GPs get annoyed a lot now. Everybody wants you to have a six-month training done in something. So how come we can do six months in breast cancer, six months in cervical cancer, six months in this cancer. So, it’s getting a bit cliché as well, like GPs are in the best position to have it so we usually laugh about it. So, I think if you publish your guidelines like GPs are best suited for that, we’re not even going to look at it…*
(GP-3)

### 4.3. Clinical Practices Relating to Oral Cancer

A wide variation appeared in the clinical practices of GPs regarding oral cancer.

#### 4.3.1. Routine Check-Ups and Examinations

All GPs recalled asking about smoking and alcohol while taking medical history; however, only a handful (*n* = 5) admitted to discussing areca nut products:

*We should be doing it, but we are not. We ask about, do you smoke? It’s a part of our medical profession that we have to. But it’s never been a software tick that you ask about betel use, any betel nut or any other thing*.(GP-13)

*No, not betel nut. I won’t ask betel nut, but if they are chewing something constantly any addiction, anything else, they usually say on their own, so I wouldn’t really particularly ask for betel nut as such*.(GP-11)

The majority of participants (*n* = 11) acknowledged not talking about oral cancer during routine check-ups:

*We do generally discuss the risk factors …but I don’t generally speak specifically about oral cancer, no*.(GP-10)

Seven GPs admitted conducting mouth/oral check-ups at some stage of routine examination only if necessary:

*Oral cancer is not—it’s not something I routinely check for unless someone is a smoker and then I will do the check, I’ll ask them some questions about any sore throat or hoarseness of their voice or any mass lesion or things about that but, if they don’t smoke then I usually don’t do it; don’t screen for it*.(GP-9)

#### 4.3.2. Referral Processes

No clear or consistent oral cancer-referral pathways were followed by interviewees. The majority of GPs (*n* = 9) were found to refer suspected oral cancer cases to an oral surgeon, while others (*n* = 5) preferred ENT (Ear, Nose, and Throat) specialists:


*There is no specialist pathway that’s what you do if you think it’s oral cancer. I would probably send him to a centre…*
(GP-12)

Few participants (*n* = 3) identified dentists as their main referral choice for oral cancer diagnosis:


*It really depends on what the lesion is…if it’s more on the gums or on the—yeah more on the gums area then I would just refer to a dentist…*
(GP-4)

#### 4.3.3. Preventative Counselling

Almost all participants raised the issue of not having any educational resource for oral cancer counselling. Although many (*n* = 9) accessed the internet, three GPs cited the ‘UpToDate’ website as a useful online resource:

*It’s also we basically use the guidelines, therapeutic guidelines for oral cancer, but there’s no like true resources, apart from online…I don’t have any access to up to date and other such data. I wish, I’d like to, but I don’t have, no*.(GP-13)

No one confirmed if any kind of standard screening tool is available for oral cancer risk assessment:


*Not particularly oral cancer, as there is no screening program available in Australia for oral cancer. Generally, it’s dealt as one of the risks that you get from smoking. So, you just touch base…*
(GP-1)

Most interviewees (*n* = 10) were keen to know more about a quick oral cancer risk evaluation guide or tool for clinical examination.

On the other hand, three GPs expressed an interest in testing an assessment tool prior to employing it in clinical practice:

*I’d be interested to see how that [assessment tool] pans out. I’d be happy to give my comments on how usable it is in general practice*.(GP-8)

## 5. Discussion

This is the first study to explore the oral cancer-related knowledge, beliefs, and clinical practices of GPs in Australia, with a particular focus on the oral cancer risk behaviours of Indian immigrants. The diverse sample of participants recruited in terms of experience, ethnicity, professional training, and location of practice has provided a valuable insight into this under-researched yet emerging area of cancer care in Australia.

The findings revealed that GPs were well aware of the key factors contributing to oral cancer, including some of the common risk products such as tobacco and alcohol. However, their knowledge regarding the association of oral cancer with areca (betel) nut consumption and the availability of these preparations was variable. The lack of awareness among some GPs in this study regarding areca nut as an oral cancer risk factor has been observed in the UK and the USA, where there has been a long history of migration from the Indian subcontinent [49,50,51]. A recent systematic review [34] on this topic found variable knowledge (0.8–50%) of GPs on emerging risk factors, such as betel nut chewing. This finding is not surprising, as the current undergraduate medical curriculum and CPD programs in Australia [34] have limited focus on oral cancer and related emerging risk factors. This limitation is evident in other developed countries as well [52,53,54]. However, many developing countries [55,56,57] have placed more emphasis in this area through well-formulated training strategies due to the alarming rise in oral cancer cases. This was evident in our study findings, as the GPs who trained overseas in India and Pakistan were well-informed about oral cancer aetiology due to better access to training modules or prior undergraduate training [58,59]. It is usual for medical curricula to reflect current health trends and challenges in the country, but in light of increasing immigration from India [32] and reports of areca (betel) nut use becoming popular in Australia [28,29,60], it is important that medical students in Australia should be made aware of these new carcinogens [61]. Needless to say, the inclusion of new emerging oral cancer risk factors in undergraduate medical curriculum [5] and continuing relevant education courses and training modules [62] for GPs would be beneficial for the early identification of oral premalignant or malignant lesions.

This study identified overall positive beliefs of GPs regarding their vital role in oral cancer prevention. However, their attitude on the current oral cancer scenario in Australia was varied, as more than half of the GPs did not consider it to be an emerging health problem. Although oral cancer has not gained much attention in Australia because of its low prevalence, recent data [26,27] indicate that oral cancer cases have increased, which interestingly aligns with a significant jump in immigration from India. Furthermore, GPs in this study who had dealt with Indian patients previously did acknowledge the continuation of typical risk habits such as betel quid chewing of Indian immigrants and thus were comparatively more concerned with this growing problem in Australia. These findings align with past studies that show the changing trends in oral cancer cases in response to cultural risk practices of Indian immigrants in other major immigration nations such as the United States, the UK, and Canada [19]. However, limited research has been undertaken to assess the oral cancer risk susceptibility of Indian communities in Australia, which may be the reason why many GPs were uncertain of oral cancer being an emerging health challenge in Australia. Mixed perceptions of GPs concerning the healthcare choices of Indian immigrants and the inclination of Indians towards healthcare providers of a similar cultural background in Australia was also noteworthy. The literature suggests that cultural factors influence beliefs, behaviours, perceptions, and emotions, all of which affect health and healthcare [63]. Therefore, the cultural background of GPs may play a role in how oral cancer risk is perceived. Culturally relevant counselling in primary care settings and the education of Indian immigrants can be useful here, as acceptance and satisfaction are high if the patient has been previously educated about oral cancer [64].

Limited knowledge of GPs regarding new oral cancer risk factors and varied beliefs towards the oral cancer scenario in Australia seems to have influenced their clinical practices in this area. The majority of GPs acknowledged rarely conducting oral cancer screening unless patients raised oral health concerns. Furthermore, there was variation in the preventative oral cancer counselling strategies among GPs with some hesitation about routinely discussing this topic with patients. This observation was expected given that GPs face many barriers including time constraints, lack of prior knowledge on this subject, and a paucity of oral cancer-specific resources for educating patients. Similar behaviours of GPs were observed in a previous review [34] that linked inadequate clinical oral cancer screening and counselling practices of GPs to their insufficient training in this area and limitations in terms of time for medical visits. The discomfort expressed by some GPs to initiate oral cancer linked discussions with patients could be one of the contributing factors in delayed diagnoses. This compliments the interpretations by Vogt et al. [65] that the hesitation of GPs about facilitating discussions about risk factors with routine patients can influence the diagnostic judgements.

The additional challenges for GPs as evidenced in the findings are the unavailability of an appropriate oral cancer screening tool and the lack of clear referral pathway to specialist services. These findings can be concerning, as earlier studies have described that gaps in oral cancer referral systems [66,67] are responsible for delays in the diagnosis. In a similar context, the absence of a national oral cancer screening program in the Australian healthcare system also adds up to the list of concerns [68]. This study supports the recommendation by Farah et al. [25] for opportunistic oral cancer screening of at-risk populations in Australia with the focus on risky health behaviours. Likewise, it is now vital for GPs serving large Indian populations to play an active role in delivering oral cancer information, after a recent study [35] has verified the involvement of Indian immigrants in risk habits in Australia.

It is also important to point out that communication about oral cancer risk factors can be complicated for both the clinician and the patient owing to many factors including health literacy, language, and cultural barriers. Additionally, sometimes, despite understanding the risk factors, clinicians find it hard to convey this information to patients from CALD (Culturally and Linguistically Diverse) backgrounds. These issues raise the need for appropriate educational resources on emerging oral cancer risk factors not only for health professionals but also for the general population, which could be more informative if translated into key languages. A recent initiative by a state government in Australia to publish an online resource for health professionals about hazardous health effects associated with areca (betel) nut chewing among immigrant populations is a step in the right direction to increase health literacy in this area [69].

The findings from this study have significant implications for oral cancer-related research, policies, and clinical practices in Australia. Further large-scale quantitative research is fundamental to confirm the findings from this study by exploring the knowledge and clinical practices of all Australian GPs in this area. A formal targeted demographic pre-selection of participants in relation to the geographic distribution of various types of oral/head and neck cancer might help in obtaining more generalisable information in future research. From a policy point of view, the formulation of clinical guidelines is essential to ensure a single national strategy for oral cancer awareness and prevention. The inclusion of a qualified dental specialist accompanying GPs during oral cancer screening and check-ups could also be beneficial and should be investigated further. However, it is important to point out that although such shared models of care exist in other countries such as the UK [70], it will be more challenging in Australia due to the lack of a universal dental scheme. Additionally, the implementation of a system and/or protocols for screening while taking into account the different pathologies according to age could be helpful in the timely identification of oral cancer/lesions [71]. GPs may benefit from the inclusion of culturally appropriate risk factors in the medical curriculum for a better understanding of oral cancer causative factors. While the overload on general practice is explicit and understandable, it is possible that accessible oral cancer-related training such as short online learning resources and continuing education courses designed for GPs could be undertaken to aid their consultation process for oral cancer prevention [72]. These resources should focus more on new oral cancer risk factors such as areca (betel) nut use, which is required considering the changing global migration and oral cancer trends.

This study also implies that GPs could adopt patient-specific decision-making strategies in their clinical practices for the identification of oral cancer through opportunistic screening of high-risk populations such as Indian immigrants who are known to be engaged in tobacco and/or alcohol consumption or chew betel/areca nut [73,74]. Being the primary contact for accessing healthcare, it becomes more crucial for GPs to engage in duties of oral cancer prevention counselling, initial screening, and routine oral/mouth check-ups [75]. Moreover, as per the practice feasibility, one-to-one health advice by GPs to high-risk populations through preventive counselling can be very effective if tailored to individual cultures and circumstances.

The timely access to medical and dental facilities has become more critical for the general population under the current COVID-19 pandemic and could lead to oral cancer being a ticking time bomb in Australia [76,77]. This situation might be more exacerbated for Indian immigrants who are less frequent in their routine medical and/or dental visits in Australia [35] and, in general, have a limited understanding for the concept of screening healthy individuals [19]. Given the great influx of Indian immigrants in recent years, the need of the hour is to develop an evidence-based oral cancer awareness resource to support public health messaging in Australia, which currently does not exist. Additionally, the development of an appropriate screening tool and a clear referral pathway could help in early diagnosis and avoid delays in initialising timely treatment.

This study has some limitations. First and foremost, the GPs were mainly from NSW, and therefore, the findings may not be representative of the GP community in VIC and other parts of Australia. Future in-depth research is needed to understand the perspectives of GPs practicing in other states and territories in Australia. Given that some of the participating GPs were trained overseas, their responses regarding oral cancer awareness may have been influenced by their previous experiences; thus, the results reported here are subject to information bias. Another limitation could be ‘volunteer error’, as some of the GP participants would have participated because they are highly motivated to learn more about the cancer problem and help their patients. Despite these limitations, this study has provided valuable insights into this under-investigated area in Australia.

## 6. Conclusions

This study has identified that GPs have good oral cancer-related knowledge with limited information about emerging risk factors, varied beliefs about the seriousness of oral cancer in Australia, and inconsistent clinical practices relating to routine oral cancer check-ups and screening as well as dental referrals. GPs play a crucial role as primary healthcare providers and are a gateway to access specialist services; however, the lack of relevant training is making it difficult for them to actively promote oral cancer prevention. Further research is warranted to confirm these findings and inform about the development of resources and/or training aimed at medical practitioners to raise awareness of oral cancer among high-risk populations such as Indian immigrants in Australia.

## Figures and Tables

**Table 1 ijerph-18-11111-t001:** Demographics of participants.

Age Range/Group	Gender	Country of Birth	Region/State	Medical Qualifications (Country)	Years of Clinical Practice in Australia
25–34	Female	Pakistan	South Sydney/NSW	Undergraduate (Pakistan) Fellowship (Australia)	3
35–44	Male	India	North Melbourne/VIC	Undergraduate (Russia) Post-graduate (Russia)	4
35–44	Male	Pakistan	South Sydney/NSW	Undergraduate (Pakistan) Fellowship (Australia)	6
35–44	Female	Philippines	South-West Melbourne/VIC	Post-graduate (Philippines) Fellowship (Australia)	4
25–34	Female	Australia	South Sydney/NSW	Undergraduate (Australia) Post-graduate (Australia)	3
55–64	Female	India	Western Sydney/NSW	Undergraduate (India) Fellowship (Australia)	23
55–64	Male	Afghanistan	Western Sydney /NSW	Undergraduate (Afghanistan) Post-graduate (Afghanistan)	20
55–64	Male	Malaysia	South-West Sydney/NSW	Undergraduate (Australia)	35
35–44	Male	Australia	South Sydney/NSW	Undergraduate, Fellowship (Australia)	19
25–34	Male	Australia	South-East Sydney/NSW	Undergraduate, Fellowship (Australia)	3
25–34	Female	India	Western Sydney/NSW	Undergraduate, Fellowship (Australia)	3
45–54	Male	India	Sydney-East/NSW	Undergraduate (India) Fellowship (UK)	6
35–44	Male	Pakistan	North-West Sydney/NSW	Undergraduate (Pakistan) Fellowship (Australia)	16
35–44	Male	India	North-West Sydney/NSW	Undergraduate (India) Fellowship (UK)	5

**Table 2 ijerph-18-11111-t002:** Categories and subcategories.

Categories	Subcategories
Oral cancer-related knowledge	Recognition of the symptoms and risk factors Availability of risk products Oral cancer training
Oral cancer-related beliefs	Views towards oral cancer scenario in Australia Perceived role in oral cancer prevention Overcoming the barriers to oral cancer prevention and diagnosis
Clinical practices relating to oral cancer	Routine check-ups and examinations Referral processes Preventative counselling

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
