# Peer review of "Perceptions and Practices of General Practitioners towards Oral Cancer and Emerging Risk Factors among Indian Immigrants in Australia: A Qualitative Study"

_ijerph, 2021, doi:10.3390/ijerph182111111_

Round 1

Reviewer 1 Report

Given the large ethno-cultural variation in the population of Australia, with large numbers of Asians of varying descent, in both the providers of healthcare and potential patients, a study of this type seems very appropriate.  Whatever information is provided from the author correctly naming it a qualitative study with much quantitative information from fourteen out of an unknown number of possible participants is obviously limited by the small sample. Clearly, the detailed results from this limited group of respondents have to be considered as being subject to a ‘volunteer error’; that is, those who participate are highly motivated to do so, usually to refer for treatment and to help the patient population in their respective catchment areas learn more about the cancer problem. Perhaps the practitioners are personally interested in learning more about the medical and public health aspects of the problem as well.  However, the reviewer is not really able to get a clear idea of the training and evaluation of the GPs in Australia and how representative this sub-sample is in relation to the geographic distribution of the various types of oral and head and neck cancer. Perhaps a more formal demographic pre-selection of the respondents might have helped obtain more generalizable information. On the positive side, the problem has certainly aroused enough interest in a sub-sample of the practicing GPs to generate some CE courses.

With respect to the number of cancers of the oral cavity, there is no mention of HPV-initiated oral cancers in males, which may well have a cultural oral sex component. Relative to barriers, communication about the risk factors suffers from the world-wide health literacy problem of both the clinician and the patient. It is sometimes difficult for clinicians to have enough understanding and ability to explain risk factors in a sufficient manner to translate to the patients to truly understand the significance of the numbers about the population in general  and how they have specific reference to themselves.

As an overall assessment of the importance of this well-done but limited study, it is certainly information which should get out to the professional and ultimately to the lay populations, but it is really not appropriate to publish it as the results of a formal research project qualitatively or quantitatively. Perhaps it would be more appropriately disseminated as a commentary. For the reviewer, it was also surprising that there was no reference at all about the role of dentists in at least the recognition and diagnosis of oral cancer.

Author Response

Thank you for providing us the opportunity to revise our manuscript for International Journal of Environmental Research and Public Health. Please find below the details of changes we have amended in the light of the reviewers’ comments. All changes in response to reviewer’s comments are indicated with track changes in the manuscript. It should be noted that some line numbers (after change in each page) are missing in the manuscript version with revisions (unfortunately couldn’t be corrected in this format due to track changes- the word document tries to be consistent with the line numbers even the ones which have been deleted). Therefore, the revisions have been addressed according to the page and line numbers in the current format (with track changes).

The following revisions have been made to the manuscript following reviewers’ comments:

REVIEWER 1:

Comment 1: Given the large ethno-cultural variation in the population of Australia, with large numbers of Asians of varying descent, in both the providers of healthcare and potential patients, a study of this type seems very appropriate.

Authors’ response: Thank you for your positive remark.  

Comment 2: Whatever information is provided from the author correctly naming it a qualitative study with much quantitative information from fourteen out of an unknown number of possible participants is obviously limited by the small sample.

Authors’ response: Thank you for the comment. There’s been limited information about the oral cancer-related knowledge and clinical practices of GPs (General Practitioners) in Australia, especially around emerging oral cancer risk practices among new immigrants (Saraswat, N., et al., Knowledge, attitudes and practices of general medical practitioners in developed countries regarding oral cancer: an integrative review. Family practice, 2020. 37(5): p. 592-605). Exploratory qualitative research has been identified as a suitable method to study areas (specifically within healthcare practice) that have previously received little or no attention (Hunter, D., McCallum, J. and Howes, D. (2019) Defining Exploratory-Descriptive Qualitative (EDQ) research and considering its application to healthcare. Journal of Nursing and Health Care, 4(1)). Therefore, due to lack of information in this area, undertaking an initial exploratory study seemed appropriate to scope of the study topic and to inform larger quantitively studies in the future. We have reiterated this information under section Materials and Methods (subsection- Design) in the paper. (Pg. 3, line 188-191)

Regarding the comment about limited sample size, we believe fourteen participants was sufficient for an exploratory study particularly as we reached data saturation and had diversity in the sample through purposive sampling. Previous studies have recommended that a minimum sample size of at least 12 is sufficient to reach data saturation in qualitative studies (Fugard AJ, Potts HW. Supporting thinking on sample sizes for thematic analyses: a quantitative tool. Int J Soc Res Methodol. 2015;18(6):669–84.; Weller, S.C., et al., Open-ended interview questions and saturation. PloS one, 2018. 13(6): p. e0198606.). Further, a recent systematic review of interview-based qualitative studies found eight to ten participants was appropriate in studies that were exploratory in nature (Vasileiou, K.; Barnett, J.; Thorpe, S.; Young, T. Characterising and justifying sample size sufficiency in interview-based studies: Systematic analysis of qualitative health research over a 15-year period. BMC Med. Res. Methodol. 2018, 18, 148). Furthermore, the sample size of the existing qualitative studies that have explored the oral cancer-related knowledge, awareness and/or practices of GP’s have ranged from 2-19 (Morse DE, Vélez Vega CM, Psoter WJ et al. Perspectives of San Juan healthcare practitioners on the detection deficit in oral premalignant and early cancers in Puerto Rico: a qualitative research study. BMC Public Health 2011; 11: 391; Cruz GD, Shulman LC, Kumar JV, Salazar CR. The cultural and social context of oral and pharyngeal cancer risk and control among Hispanics in New York. J Health Care Poor Underserved 2007; 18: 833–46; Canto MT, Horowitz AM, Child WL. Views of oral cancer prevention and early detection: Maryland physicians. Oral Oncol 2002; 38: 373–7).These points have been reiterated in the data collection section of the paper. (Pg. 4, line 234-237)

Comment 3: Clearly, the detailed results from this limited group of respondents have to be considered as being subject to a ‘volunteer error’; that is, those who participate are highly motivated to do so, usually to refer for treatment and to help the patient population in their respective catchment areas learn more about the cancer problem. Perhaps the practitioners are personally interested in learning more about the medical and public health aspects of the problem as well. 

Authors’ response: Thank you for the suggestion. We acknowledge that some participants can be highly motivated and personally interested in learning more about public health issues. We have inserted this as a limitation in the paper. (Pg.13, line 754, Pg. 14, line 761-763)

Comment 4: However, the reviewer is not really able to get a clear idea of the training and evaluation of the GPs in Australia and how representative this sub-sample is in relation to the geographic distribution of the various types of oral and head and neck cancer. Perhaps a more formal demographic pre-selection of the respondents might have helped obtain more generalizable information.

Authors’ response: Thank you for the comment. The aspect of training and evaluation of GPs in Australia has now been added in the sampling and recruitment section of the paper. (Pg. 4, line 219-226)

In terms of the representativeness of the sub sample, it is important to reiterate that this was an exploratory qualitative study to explore the oral cancer-related knowledge, beliefs and clinical practices of GPs in Australia with special focus on their awareness regarding new emerging factors like areca nut use among Indian immigrants.  As such it was important to ensure diversity in the sample in terms of gender, age, country of birth and educational training and this was achieved through a combination of purposive and snowball sampling, deliberately targeting suburbs in Sydney and Melbourne that were densely populated with Indian communities. We have made this point more explicit in the paper (Pg. 4, line 218). We have also included the following section comparing the demographic profile of participants with Australian national workforce. (Pg. 5, line 312-320; Pg. 6, line 327-330)

Of the 14 GPs, nine were male participants, and seven were between 35-54 years of age (range 25–64 years). More than half had obtained their basic qualifications from overseas (n=9) and had undertaken specialist general practice training in Australia (n=9). Their clinical practice experience in Australia ranged from 3 to 35 years (mean 10.7 years). (See Table 1 for demographics-given below). These demographics were fairly similar to the trends observed among GPs in Australia. Recent workforce data indicate there are more male GPs than female GPs in Australia (60% vs 40%), more than half (53%) are between the age groups of 35-54 years and the majority (51-80% depending on specialty training) have their basic qualifications from overseas (Royal Australian College of General Practitioners. General Practice- Health of the Nation. 2020; Available from: https://www.racgp.org.au/getmedia/c2c12dae-21ed-445f-8e50-530305b0520a/Health-of-the-Nation-2020-WEB.pdf.aspx.). In addition, according to the Australian Bureau of Statistics, there has been a marked increase in the number of GP’s and specialists from South Asia (Australian Bureau of Statistics. Doctors and Nurses. Australian Social Trends, April 2013 2013; Available from: https://www.abs.gov.au/ausstats/[email protected]/lookup/4102.0main+features20april+2013) and they represent the second largest group of GPs after Australian born medical practitioners (Negin, J., et al., Foreign-born health workers in Australia: an analysis of census data. Human Resources for Health, 2013. 11(1): p. 1-9)

The detailed description of the diverse sample, method and findings have helped with the transferability of the results. However, we acknowledge the findings cannot be generalised and have already mentioned this in the limitations section. We also agree a formal targeted demographic pre-selection of participants in relation to the geographic distribution of the various types of oral/head and neck cancer might help in obtaining more generalisable information in larger quantitative studies and have included this point as a recommendation for future research. (Pg. 13, line 710-712)

Comment 5: On the positive side, the problem has certainly aroused enough interest in a sub-sample of the practicing GPs to generate some CE courses.

Authors’ response: Thanks for the feedback.

Comment 6: With respect to the number of cancers of the oral cavity, there is no mention of HPV-initiated oral cancers in males, which may well have a cultural oral sex component.

Authors’ response: Thank you for this recommendation. We now have added about HPV-initiated oral cancers in the introduction section of the paper. (Pg. 2, line 107-109)

Comment 7: Relative to barriers, communication about the risk factors suffers from the world-wide health literacy problem of both the clinician and the patient. It is sometimes difficult for clinicians to have enough understanding and ability to explain risk factors in a sufficient manner to translate to the patients to truly understand the significance of the numbers about the population in general and how they have specific reference to themselves.

Authors’ response: Thank you for making this point. We agree that communication about the risk factors can be complicated for both the clinician and the patient owing to many factors including health literacy, language, and cultural barriers. We have now included some additional points regarding this in the discussion section of the paper. (Pg. 12, line 691-695, Pg. 13, line 699-704)

It is also important to point out that communication about oral cancer risk factors can be complicated for both the clinician and the patient owing to many factors including health literacy, language, and cultural barriers. Additionally, sometimes, despite understanding the risk factors, clinicians find it hard to convey this information to patients from CALD (Culturally And Linguistically Diverse) backgrounds. These issues raise the need for appropriate educational resources on emerging oral cancer risk factors, not only for health professionals but also for the general population as well which could be more informative if translated into key languages. A recent initiative by a state government in Australia to publish an online resource for health professionals about hazardous health effects associated with areca (betel) nut chewing among immigrant populations is welcome strategy to increase health literacy in this area (NSW Government. Betel nut products and preparations. Health 2021; Available from: https://www.health.nsw.gov.au/oralhealth/prevention/Pages/betel-nut-preparations-use.aspx. ).

Comment 8: As an overall assessment of the importance of this well-done but limited study, it is certainly information which should get out to the professional and ultimately to the lay populations, but it is really not appropriate to publish it as the results of a formal research project qualitatively or quantitatively. Perhaps it would be more appropriately disseminated as a commentary.

Authors’ response: Thank you for your comment. As explained earlier, this was an exploratory study undertaken to get insights into an under-researched area. An exploratory qualitative approach allows the researcher to explore a topic with limited coverage within the literature and allows the development of new knowledge in that area (Hunter, D., McCallum, J. and Howes, D. (2019) Defining Exploratory-Descriptive Qualitative (EDQ) research and considering its application to healthcare. Journal of Nursing and Health Care, 4(1)). The findings from this study if published will help inform future larger quantitative studies to confirm the findings.

Comment 9: For the reviewer, it was also surprising that there was no reference at all about the role of dentists in at least the recognition and diagnosis of oral cancer.

Authors’ response: Thanks for the comment. We do acknowledge the crucial role of dentists in diagnosis and treatment of oral cancer. However, the purpose of this study was to explore GPs’ position and potential role in oral cancer identification and prevention considering they are the first point of contact for patients to access healthcare in Australia. However, we do acknowledge that it is important to at least recognise the role of the dentist in diagnosing oral cancer and thus have provided additional information in the background section of paper. (Pg. 2, line 77-79).

Furthermore, we also have added in the discussion section about the definitive role dentists can play in oral cancer diagnoses of patients during/after initially screened by GPs. (Pg. 13, line 711-715). We have also highlighted in the paper that a recent Australian study showed that only half of all oral health professionals are performing oral cancer screening with their patients (Mariño, R., et al., Oral cancer screening practices of oral health professionals in Australia. BMC oral health, 2017. 17(1): p. 1-9). (Pg. 3, line 161-163)

Reviewer 2 Report

Overall, this is a novel and relevant manuscript for a growing subpopulation in Australia, but also relevant in south Asia. 

Because the impetus of this study seemed to be the rapid increase of a certain subpopulation, and the cultural use of areca nut chewing, more of the introduction should have focused on oral cancer and areca nut use. There is a significant amount of current, relevant evidence on this subject, including an excellent meta-analysis paper in this month's issue of Oral Diseases. However, the questions focused on oral cancer knowledge and screening and one focused on how to raise awareness/health behavior change in the high risk population, not among GP's. Why ask this question if there is already low knowledge and application of oral cancer screening among GPs? 

A mixed-methods approach would have strengthened the study design. Why was this not done? 

The N is quite small, making it difficult to generalize the findings. It would be better to present it as a pilot. Additionally, close to half of the participants were from countries where this cultural practice is high, which could skew the data. 

The conclusion does an excellent job of staying within the limitations of the study's small N. 

Author Response

Thank you for providing us the opportunity to revise our manuscript for International Journal of Environmental Research and Public Health. Please find below the details of changes we have amended in the light of the reviewers’ comments. All changes in response to reviewer’s comments are indicated with track changes in the manuscript. It should be noted that some line numbers (after change in each page) are missing in the manuscript version with revisions (unfortunately couldn’t be corrected in this format due to track changes- the word document tries to be consistent with the line numbers even the ones which have been deleted). Therefore, the revisions have been addressed according to the page and line numbers in the current format (with track changes).

The following revisions have been made to the manuscript following reviewers’ comments:

REVIEWER 2:

Comment 1: Overall, this is a novel and relevant manuscript for a growing subpopulation in Australia, but also relevant in south Asia.

Authors’ response: Thank you for this feedback.

Comment 2: Because the impetus of this study seemed to be the rapid increase of a certain subpopulation, and the cultural use of areca nut chewing, more of the introduction should have focused on oral cancer and areca nut use.  There is a significant amount of current, relevant evidence on this subject, including an excellent meta-analysis paper in this month's issue of Oral Diseases.

 Authors’ response: Thank you for the suggestion. We acknowledge that a certain aspect of our study revolves around the rapid rise of Indian immigrants in Australia and the cultural use of areca (betel) nut use among them. We now have revised Introduction to include a bit more detail on this part. The suggested paper (Yang, J., et al., Do betel quid and areca nut chewing deteriorate prognosis of oral cancer? A systematic review, meta‐analysis, and research agenda. Oral diseases, 2021. 27(6): p. 1366-1375) has been included as reference in the same section of paper. (Pg. 2, line 114-116)

Comment 3- However, the questions focused on oral cancer knowledge and screening and one focused on how to raise awareness/health behavior change in the high-risk population, not among GP's. Why ask this question if there is already low knowledge and application of oral cancer screening among GPs?

Authors’ response: Thank you for the comment. Since this area hasn’t been researched in depth in Australia till now, not much was known about GPs’ oral cancer related knowledge, beliefs and clinical practices in Australia. Our study was the first to explore this with a particular emphasis on new emerging risk factor of areca nut use which is popular among Indian immigrants. As we have mentioned in the paper this was the part of a larger mixed methods study which aimed to investigate oral cancer risk behaviours of Indian immigrants in Australia. We conducted concurrent interviews with GPs and Indian immigrants while keeping the topic/questions quite similar to explore this area. Furthermore, we tried to probe about oral cancer knowledge, screening and awareness strategies from both the GPs and Indian immigrants’ perspective. The results section of this paper (Page 9 in previous version) provides the suggestions by participants to address barriers in oral cancer prevention for both the GPs and the general population. For your reference, we have now attached the complete questionnaire (within supplementary file 2) which we used for the GP interviews to improve clarity in this section.

Comment 4- A mixed-methods approach would have strengthened the study design. Why was this not done?

Authors’ response: Thank you for your comment. As we have responded in the previous question, this study is part of a larger mixed methods project exploring oral cancer risk behaviours (knowledge, attitudes and practices) of Indian immigrants and strategies to raise awareness regarding oral cancer prevention in Australia. In the larger study a quantitative survey on this topic is being undertaken among Indian migrants in Australia (https://cohortaustralia.com/oralcancerinimmigrants/) and will be published elsewhere. Employing a mixed methods approach solely for GPs was beyond the scope of this study as it was a doctoral project. We have recommended in this paper to undertake future large-scale quantitative research to confirm these findings.

Comment 5: The N is quite small, making it difficult to generalize the findings. It would be better to present it as a pilot. Additionally, close to half of the participants were from countries where this cultural practice is high, which could skew the data.

Authors’ response: Thank you for the comment. As we have mentioned in previous comments, our aim was not to generalise the findings. This was an exploratory qualitative study to investigate an area which hasn’t received much attention in Australia till now. The detailed description of the diverse sample, method and findings have helped with the transferability of the results. However, we acknowledge the findings cannot be generalised and have already mentioned this in the limitations section.

Regarding limited sample size, we believe fourteen participants was sufficient for an exploratory study particularly as we reached data saturation and had diversity in the sample through purposive sampling. Previous studies have recommended that a minimum sample size of at least 12 is sufficient to reach data saturation in qualitative studies (Fugard AJ, Potts HW. Supporting thinking on sample sizes for thematic analyses: a quantitative tool. Int J Soc Res Methodol. 2015;18(6):669–84.; Weller, S.C., et al., Open-ended interview questions and saturation.PloS one, 2018. 13(6): p. e0198606.). Further, a recent systematic review of interview-based qualitative studies found eight to ten participants was appropriate in studies that were exploratory in nature (Vasileiou, K.; Barnett, J.; Thorpe, S.; Young, T. Characterising and justifying sample size sufficiency in interview-based studies: Systematic analysis of qualitative health research over a 15-year period. BMC Med. Res. Methodol. 2018, 18, 148). Furthermore, the sample size of the existing qualitative studies that have explored the oral cancer-related knowledge, awareness and/or practices of GP’s have ranged from 2-19 (Morse DE, Vélez Vega CM, Psoter WJ et al. Perspectives of San Juan healthcare practitioners on the detection deficit in oral premalignant and early cancers in Puerto Rico: a qualitative research study. BMC Public Health 2011; 11: 391; Cruz GD, Shulman LC, Kumar JV, Salazar CR. The cultural and social context of oral and pharyngeal cancer risk and control among Hispanics in New York. J Health Care Poor Underserved 2007; 18: 833–46; Canto MT, Horowitz AM, Child WL. Views of oral cancer prevention and early detection: Maryland physicians. Oral Oncol 2002; 38: 373–7). These points have been reiterated in the data collection section of the paper. (Pg. 4, line 234-237)

We acknowledge that five of the participants were from countries where this cultural practice is high, and we have mentioned this in the limitation. However, we have also mentioned in the demographic section that according to Australian workforce data the majority of GPs in Australia (51-80% depending on specialty training) have their basic qualifications from overseas (Royal Australian College of General Practitioners. General Practice- Health of the Nation. 2020; Available from: https://www.racgp.org.au/getmedia/c2c12dae-21ed-445f-8e50-530305b0520a/Health-of-the-Nation-2020-WEB.pdf.aspx. ). In addition, according to the Australian Bureau of Statistics, there has been a marked increase in the number of GP’s and specialists from South Asia (Australian Bureau of Statistics. Doctors and Nurses. Australian Social Trends, April 2013 2013; Available from: https://www.abs.gov.au/ausstats/[email protected]/lookup/4102.0main+features20april+2013) and they represent the second largest group of GPs after Australian born medical practitioners (Negin, J., et al., Foreign-born health workers in Australia: an analysis of census data. Human Resources for Health, 2013. 11(1): p. 1-9). (Pg. 5, line 315-318; Pg. 6, line 325-328)

Comment 6- The conclusion does an excellent job of staying within the limitations of the study's small N.

Authors’ response: Thank you for the comment.

Reviewer 3 Report

This is a potentially interesting paper, however, there are substantial flaws and several issues must be adressed before publication.

Please see attached PDF for a detailed account.

Author Response

Thank you for providing us the opportunity to revise our manuscript for International Journal of Environmental Research and Public Health. Please find below the details of changes we have amended in the light of the reviewers’ comments. All changes in response to reviewer’s comments are indicated with track changes in the manuscript. It should be noted that some line numbers (after change in each page) are missing in the manuscript version with revisions (unfortunately couldn’t be corrected in this format due to track changes- the word document tries to be consistent with the line numbers even the ones which have been deleted). Therefore, the revisions have been addressed according to the page and line numbers in the current format (with track changes).

The following revisions have been made to the manuscript following reviewers’ comments:

REVIEWER 3:

Comment 1: This is a potentially interesting paper, however, there are substantial flaws, and several issues must be addressed before publication. Please see attached PDF for a detailed account.

Authors’ response: Thank you for the feedback. We have extracted the comments from the pdf and addressed them below.

Comment 2: In Abstract:

Comment 2a: Background­­–– the background is too general for the topic at hand, is this paper discussing oral cancer in India or Australia?

Authors’ response: Thank you for the suggestion. We have now revised the background section of the abstract to focus on oral cancer in Australia and avoid confusion. (Pg. 1, line 22-24)

Comment 2b: Conclusion should be more focused and reflect the results of this paper.

Authors’ response: As suggested, conclusion section of the abstract has been refined to reflect the results of this paper. (Pg.1, line 33-36)

Comment 3: In the background section:

Comment 3a: The information is repetitive, please rephrase.

Authors’ response: Thanks for the feedback. The information has been revised in the paper. (Pg.2, line 89-91)

Comment 3b: The introduction is too long and repetitive, please shorten.

Authors’ response: We acknowledge that few sentences are repetitive in the Introduction and were making the paper long. We have now refined the Introduction section. (Pg. 2, line 97-103)

Comment 3c: Since one of the etiological factors for the appearance of oral cancer are chronic traumatic microlesions I would suggest adding a paragraph regarding wound healing. I suggest:   Martu MA, Maftei GA, Luchian I, Popa C, Filioreanu AM, Tatarciuc D, Nichitean G, Hurjui LL, Foia LG. Wound healing of periodontal and oral tissues: Part II—Patho-phisiological conditions and metabolic diseases. Rom. J. Oral Rehab. 2020 Jul;12:30-40.

Authors’ response: Thanks for the suggestion. Though wound healing is important to consider while discussion oral cancer lesion, a whole paragraph could draw attention away from the main aim of the study (exploring oral cancer-related knowledge, beliefs and practices of GPs in Australia). However, we have added a relevant sentence about wound healing from the suggested reference. (Pg. 2, line 109-111)

Comment 4: In Discussion section:

Comment 4a: the authors should discuss other oral pathologies that have interactions with oral diseases and whose treatment may cause cancer, I suggest: Martu MA, Solomon SM, Toma V, Maftei GA, Iovan A, Gamen A, Hurjui L, Rezus E, Foia L, Forna NC. The importance of cytokines in periodontal disease and rheumatoid arthritis. Review. Romanian Journal of Oral Rehabilitation. 2019 Apr;11(2):220-40.

Filioreanu AM, Popa C, Maftei GA, Parlatescu I, Nicolae CL, Popescu E. Migratory stomatitis–case presentation. Romanian Journal of Oral Rehabilitation. 2018 Oct;10(4).

Authors’ response: Thanks for the recommendation. We appreciate the reviewer’s concern regarding other oral pathologies which have interactions with oral diseases and whose treatment may cause cancer. However, we feel discussing this would provide too many details about associated pathology which is not the focus of our study/paper.

Comment 4b: This is an important aspect that relates to health policies placed in effect by governments. Authors should further discuss the necessity of implementing a system and protocols that take into account the different pathologies according to age, I suggest: Popa C, Filioreanu AM, Stelea C,Maftei GA, Popescu E. Prevalence of oral lesions modulated by patient's age: the young versus the elderly. Romanian Journal of Oral Rehabilitation. 2018 Jul 1;10(3):50-6.

Authors’ response: Thanks for the suggestion. Since oral cancers are usually diagnosed in later stages and that contributes to lesser survival rates, the implementation of a system and protocols that consider the different pathologies according to age seems like an excellent idea specifically in relation to oral cancer prevention. We have now added this as future recommendation in the paper. (Pg. 13, line 719-721)

Reviewer 4 Report

Overall recommendation: Accept after minor revisions

Brief summary: In this paper, the authors investigate the oral cancer-related knowledge, beliefs, and clinical practices of general medical practitioners (GPs) in Australia. Even though their work may significantly contribute to our field, few minor revisions are needed before consideration for publication.

Main comments

  1. The sample is quite small (No of participants: 14); similar studies have considerably larger (e.g. “Riordain, R.N. and C. McCreary, Oral cancer–Current knowledge, practices and implications for training among an Irish general 617 medical practitioner cohort. Oral oncology, 2009. 45(11): p. 958-962.”; 236 questionnaires available for analysis).
  2. It would be better if the consent of all the participants was written and not verbal. Furthermore, the return of the transcripts back to the participants would be also preferred.
  3. The participants could be also asked for their knowledge regarding oral potentially malignant disorders (OPMDs; leukoplakia, erythroplakia, erythroleukoplakia, etc.).
  4. Please provide an indicative questionnaire that was used in these semi-structured interviews (at least the standard questions).
  5. Please provide tables summarizing the participants’ answers regarding risk factors, signs and symptoms, referrals, attendance in oral cancer CME courses, formal teaching on oral cancer (diagnosis, clinical examination, referral etc.) during training, etc.
  6. The specific age of each participant could be also provided before their categorization.
  7. Line No 211: “Recognition of symptoms…” is not reported. “Non-healing ulcer” and “lump” both represent clinical findings/signs. However, symptoms (e.g. bleeding, pain, halitosis) could be also reported.
  8. The citation of a paper that is not published yet - although being under review - may not be acceptable.
  9. “Lechner, M., et al., A cross-sectional survey of awareness of human papillomavirus-associated oropharyngeal cancers among general 615 practitioners in the UK. BMJ open, 2018. 8(7): p. e023339.” (No 33 on references) refers to oropharyngeal and not oral cancer - a pathogenically different entity - and therefore it should be replaced by another one.
  10. For oral cancer new cases and statistics it would be more acceptable to use GLOBOCAN (e.g. “Sung H., Ferlay J., Siegel R.L., Laversanne M., Soerjomataram I., Jemal A., Bray F. Global cancer statistics 2020: GLOBOCAN estimates of incidence and mortality worldwide for 36 cancers in 185 countries. CA Cancer J. Clin. 2021 doi: 10.3322/caac.21660.”).
  11. “Sarumathi, T., et al., Awareness and knowledge of common oral diseases among primary care physicians. Journal of clinical and 553 diagnostic research: JCDR, 2013. 7(4): p. 768.” (No 3 on references) does not refer to AIDS and diabetes.
  12. On line No 94 the authors provide specific data that could not be confirmed on “Department of Home Affairs. 2018 – 19 Migration Program Report. 2019 [cited 2021 6th March]; Available from: 595 https://www.homeaffairs.gov.au/research-and-stats/files/report-migration-program-2018-19.pdf.” (No 24 on references).

Additional minor comments

  1. Participants’ phrases literally cited should be on “…”.
  2. On the abstract, please report the full term (“general medical practitioners”) before the use of GPs on line No 23.
  3. Please replace the word “inspection” on line No 55 with “examination”.
  4. Please replace the word “from” on line No 129 with “of”.
  5. Please replace the word “GP’s” on lines No 459 and 472 with “GPs”.
  6. The authors could occasionally replace the word “explore” with “investigate” to avoid repetition.
  7. Please replace the word “Florian” on line 469 with “Vogt”.
  8. Please avoid repetition of the phrase “in Australia” on lines No 447-448.
  9. Please replace the word “was” on line 451 with “were”.
  10. Please replace the word “Up-to-date” on lines 390-391 with “UpToDate”.
  11. Please replace the word “lump in the mouth” on line No 231 with “oral swelling”.
  12. The word “also” on line No 107 could be omitted.
  13. Please decide on whether you will use “serial/Oxford commas” or not. Although this grammar “technique” is not generally used in the manuscript, it “appears” on line No 105.
  14. Line No 60: *100,000
  15. Line 141: “(see supplementary file 2 space - Interview focus areas)”
  16. Line No 173: “study backspace.”.
  17. Lines No 201-202: *parti-cipants.
  18. Lines No 213-214: “like tobacco and alcohol”.
  19. Line No 252: “quote:”.
  20. Line No 351: “Because”.
  21. Line No 370: “We”.
  22. Lines No 421-422: transfer the reference “[42]” at the end of the sentence.

Author Response

Thank you for providing us the opportunity to revise our manuscript for International Journal of Environmental Research and Public Health. Please find below the details of changes we have amended in the light of the reviewers’ comments. All changes in response to reviewer’s comments are indicated with track changes in the manuscript. It should be noted that some line numbers (after change in each page) are missing in the manuscript version with revisions (unfortunately couldn’t be corrected in this format due to track changes- the word document tries to be consistent with the line numbers even the ones which have been deleted). Therefore, the revisions have been addressed according to the page and line numbers in the current format (with track changes).

The following revisions have been made to the manuscript following reviewers’ comments:

REVIEWER 4:

Comment 1: Overall recommendation: Accept after minor revisions.

Authors’ response: Thanks for the feedback.

Comment 2: Brief summary: In this paper, the authors investigate the oral cancer-related knowledge, beliefs, and clinical practices of general medical practitioners (GPs) in Australia. Even though their work may significantly contribute to our field, few minor revisions are needed before consideration for publication.

Authors’ response: Thanks for your comment.

Comment 3: Main comments-

Authors’ response: As suggested, the main comments have been addressed as sub-sections/sub-questions given below.

Comment 3a: The sample is quite small (No of participants: 14); similar studies have considerably larger (e.g. “Riordain, R.N. and C. McCreary, Oral cancer–Current knowledge, practices and implications for training among an Irish general 617 medical practitioner cohort. Oral oncology, 2009. 45(11): p. 958-962.”; 236 questionnaires available for analysis).

Authors’ response: Thanks for the comment. As we have mentioned before in the response letter, this was an exploratory qualitative study to investigate an under researched area- GPs’ oral cancer related knowledge and practices in Australia. We believe fourteen participants was sufficient for an exploratory study particularly as we reached data saturation and had diversity in the sample through purposive sampling. Previous studies have recommended that a minimum sample size of at least 12 is sufficient to reach data saturation in qualitative studies (Fugard AJ, Potts HW. Supporting thinking on sample sizes for thematic analyses: a quantitative tool. Int J Soc Res Methodol. 2015;18(6):669–84.; Weller, S.C., et al., Open-ended interview questions and saturation. PloS one, 2018. 13(6): p. e0198606.). Further, a recent systematic review of interview-based qualitative studies found eight to ten participants was appropriate in studies that were exploratory in nature (Vasileiou, K.; Barnett, J.; Thorpe, S.; Young, T. Characterising and justifying sample size sufficiency in interview-based studies: Systematic analysis of qualitative health research over a 15-year period. BMC Med. Res. Methodol. 2018, 18, 148). Furthermore, the sample size of the existing qualitative studies that have explored the oral cancer-related knowledge, awareness and/or practices of GP’s have ranged from 2-19 (Morse DE, Vélez Vega CM, Psoter WJ et al. Perspectives of San Juan healthcare practitioners on the detection deficit in oral premalignant and early cancers in Puerto Rico: a qualitative research study. BMC Public Health 2011; 11: 391; Cruz GD, Shulman LC, Kumar JV, Salazar CR. The cultural and social context of oral and pharyngeal cancer risk and control among Hispanics in New York. J Health Care Poor Underserved 2007; 18: 833–46; Canto MT, Horowitz AM, Child WL. Views of oral cancer prevention and early detection: Maryland physicians. Oral Oncol 2002; 38: 373–7). These points have been reiterated in the data collection section of the paper. (Pg. 4, line 234-237)

The example given by the reviewer (Riordain, R.N. and C. McCreary, Oral cancer–Current knowledge, practices and implications for training among an Irish general 617 medical practitioner cohort. Oral oncology, 2009. 45(11): p. 958-962) was a quantitative study involving a survey. Although the suggested study is very relevant to this subject area, we feel it cannot be used as a comparison as it had a different methodological approach.

Comment 3b: It would be better if the consent of all the participants was written and not verbal. Furthermore, the return of the transcripts back to the participants would be also preferred.

Authors’ response: We acknowledge the comment regarding consent. It is well documented that recruiting GPs and undertaking data collection is very challenging. Due to time constraints and busy schedules of GPs, conducting interviews over the phone was the only practical way of collecting data as it offered participants flexibility. Nearly all interviews (n=13) have been conducted over phone at the time convenient for GPs (between the patient appointments, while driving and having lunch). Furthermore, we received ethics approval to obtain written or verbal (informed) consent depending on what was more feasible for GP participants. Regarding transcripts, participants were verbally asked at the time of interviews if they were interested to review them but none of them agreed due to their busy schedules. This was mentioned in the manuscript (data collection section)- “The transcripts were not returned for member checking due to time constraints of participants.”

Comment 3c: The participants could be also asked for their knowledge regarding oral potentially malignant disorders (OPMDs; leukoplakia, erythroplakia, erythroleukoplakia, etc.).

Authors’ response: Thanks for the comment. Oral pre-malignant lesions/disorders are definitely important in oral cancer identification process. However, it was too detailed/in-depth for the study interviews which were focused on basic knowledge about oral cancer. In addition, time constraint was also a factor given the overly burdened clinical routine of GPs.

This aspect can be explored in future quantitative work (as an item in questionnaire/survey) and would add valuable information to existing literature.

Comment 3d: Please provide an indicative questionnaire that was used in these semi-structured interviews (at least the standard questions).

Authors’ response: Thanks for raising this point. We did provide the interview topic guide depicting the areas which were explored through this semi structured interviews. We have now attached more detailed questionnaire (within supplementary file 2) that was used to initiate the discussion and probe further depending on participants’ response.

Comment 3e: Please provide tables summarizing the participants’ answers regarding risk factors, signs and symptoms, referrals, attendance in oral cancer CME courses, formal teaching on oral cancer (diagnosis, clinical examination, referral etc.) during training, etc.

Authors’ response: Thanks for the comment. Since our paper is based on findings from an exploratory qualitative study, response from the participants across the semi structured interviews were varied and not all the participants discussed every topic area during interviews (owing to various factors). Hence it wasn’t possible to summarise/categorise these findings in tables.

Comment 3f: The specific age of each participant could be also provided before their categorization.

Authors’ response: Thanks for the comment. We asked age range at time of interviews with a purpose to categorise the participants in specific age groups. Therefore, we don’t have the actual age of each participant to provide in the paper. However, we have now collapsed the age groups/range in demographics table (Pg.6) to provide better comparability with the available latest statistics regarding GPs in Australia (Royal Australian College of General Practitioners. General Practice- Health of the Nation. 2020; Available from: https://www.racgp.org.au/getmedia/c2c12dae-21ed-445f-8e50-530305b0520a/Health-of-the-Nation-2020-WEB.pdf.aspx. ). The details are also provided in the paper. (Pg. 5, line 315-318)

Comment 3g: Line No 211: “Recognition of symptoms…” is not reported. “Non-healing ulcer” and “lump” both represent clinical findings/signs. However, symptoms (e.g. bleeding, pain, halitosis) could be also reported.

Authors’ response: Thanks for the feedback. We have now revised the results section as per the findings regarding the symptoms for oral cancer. (Pg. 7, line 388-391)

Comment 3h: The citation of a paper that is not published yet - although being under review - may not be acceptable.

Authors’ response: Thanks for the comment. We acknowledge the cited paper is still under review. The paper has been resubmitted with minor revisions and will be accepted shortly. However, for the purpose of this paper we have removed this citation. (Removed From introduction and data collection section)

Comment 3i: “Lechner, M., et al., A cross-sectional survey of awareness of human papillomavirus-associated oropharyngeal cancers among general 615 practitioners in the UK. BMJ open, 2018. 8(7): p. e023339.” (No 33 on references) refers to oropharyngeal and not oral cancer - a pathogenically different entity - and therefore it should be replaced by another one.

Authors’ response: Thank you for highlighting this. This paper (Lechner, M., et al., A cross-sectional survey of awareness of human papillomavirus-associated oropharyngeal cancers among general 615 practitioners in the UK. BMJ open, 2018. 8(7): p. e023339) was cited in the discussion section with the purpose to present lack of awareness among GPs regarding harmful consequences areca nut use. However, acknowledging the reviewer’s concern, we now have removed this reference. (Pg. 11, line 617)

Comment 3j: For oral cancer new cases and statistics it would be more acceptable to use GLOBOCAN (e.g. “Sung H., Ferlay J., Siegel R.L., Laversanne M., Soerjomataram I., Jemal A., Bray F. Global cancer statistics 2020: GLOBOCAN estimates of incidence and mortality worldwide for 36 cancers in 185 countries. CA Cancer J. Clin. 2021 doi: 10.3322/caac.21660.”).

Authors’ response: Thank you for the suggestion. We agree with the reviewer and have added the suggested reference as required in the paper. (Pg. 2, line 91)

Comment 3k: “Sarumathi, T., et al., Awareness and knowledge of common oral diseases among primary care physicians. Journal of clinical and 553 diagnostic research: JCDR, 2013. 7(4): p. 768.” (No 3 on references) does not refer to AIDS and diabetes.

Authors’ response: Thanks for highlighting this. We acknowledge that the reference mentioned by reviewer has used broad term ‘oral diseases’ and does not specifically refer to AIDS and diabetes. We now have replaced this reference with other one (Petersen, P.E., et al., The global burden of oral diseases and risks to oral health.Bulletin of the World Health Organization, 2005. 83: p. 661-669) and have further refined the sentence in the introduction section. (Pg.2, line 72-73)

Comment 3l: On line No 94 the authors provide specific data that could not be confirmed on “Department of Home Affairs. 2018 – 19 Migration Program Report. 2019 [cited 2021 6th March]; Available from: 595 https://www.homeaffairs.gov.au/research-and-stats/files/report-migration-program-2018-19.pdf.” (No 24 on references).

Authors’ response: Thanks for picking this up. We now have updated the reference (Department of Home Affairs. 2019 – 20 Migration Program Report. 2020; Available from: https://www.homeaffairs.gov.au/research-and-stats/files/report-migration-program-2019-20.pdf) in paper (Pg. 3, line 157). The specific data can be found on the given link (Page no. 9 and Page 12 of the report).

Comment 4: Additional minor comments-

Authors’ response: As suggested, the main comments have been addressed as sub-sections/sub-questions given below.

Comment 4a: Participants’ phrases literally cited should be on “…”.

Authors’ response: Thanks for the suggestion. We adopted the style to depict quotes in italics as per journal requirements and therefore didn’t cite in “ ”.

Comment 4b: On the abstract, please report the full term (“general medical practitioners”) before the use of GPs on line No 23.

Authors’ response: Thanks. The suggested correction has been made in the abstract. (Pg. 1, line 24)

Comment 4c: Please replace the word “inspection” on line No 55 with “examination”.

Authors’ response: As suggested, the word “inspection” has been replaced with “examination”. (Pg. 2, line 85)

Comment 4d: Please replace the word “from” on line No 129 with “of”.

Authors’ response: The suggested correction has been made in the paper. (Pg. 4, line 209)

Comment 4e: Please replace the word “GP’s” on lines No 459 and 472 with “GPs”.

Authors’ response: Thanks. The word “GP’s” has been replaced with GPs as required. (Pg. 12, line 669/682)

Comment 4f: The authors could occasionally replace the word “explore” with “investigate” to avoid repetition.

Authors’ response: Thanks for the suggestion, we have now replaced word “explore” with “investigate” at few places to avoid repetition. (Pg. 3, line 174; Pg.4, line 208; Pg. 14, line 763)

Comment 4g: Please replace the word “Florian” on line 469 with “Vogt”.

Authors’ response: As suggested, the word “Florian” has been replaced with “Vogt”. (Pg.12, line 679)

Comment 4h: Please avoid repetition of the phrase “in Australia” on lines No 447-448.

Authors’ response: Thanks for picking this up. We have removed phrase “in Australia” on the mentioned line. (Pg. 11, line 612)

Comment 4i: Please replace the word “was” on line 451 with “were”.

Authors’ response: As suggested, the word “was” now has been replaced by “were”. (Pg.11 , line 614)

Comment 4j: Please replace the word “Up-to-date” on lines 390-391 with “UpToDate”.

Authors’ response: Thanks for the suggestion. We have replaced the word “Up-to-date” with “UpToDate”. (Pg.11, line 587)

Comment 4k: Please replace the word “lump in the mouth” on line No 231 with “oral swelling”.

Authors’ response: As suggested, the word “lump in the mouth” has been replaced with “oral swelling”. (Pg. 7, line 389)

Comment 4l: The word “also” on line No 107 could be omitted.

Authors’ response: Thanks. We agree with the reviewer and have removed the word “also” from the sentence. (Pg. 3, line 174)

Comment 4m: Please decide on whether you will use “serial/Oxford commas” or not. Although this grammar “technique” is not generally used in the manuscript, it “appears” on line No 105.

Authors’ response: Thanks for making this point. We have decided not to use serial commas in the manuscript and have revised paper accordingly.

Comment 4n: Line No 60: *100,000

Authors’ response: Thanks. We have corrected this in paper. (Pg. 2, line 91)

Comment 4o: Line 141: “(see supplementary file 2 space - Interview focus areas)”

Authors’ response: Thanks for picking this up. We have added the space as required, in the paper. (Pg. 4, line 231)

Comment 4p: Line No 173: “study backspace.”

Authors’ response: Thanks. The extra space has been removed. (Pg.5, line 281)

Comment 4q: Lines No 201-202: *parti-cipants.

Authors’ response: Thanks for noticing this. We have corrected it in the paper.

Comment 4r: Lines No 213-214: “like tobacco and alcohol”.

Authors’ response: As suggested, we have added “and” as required. (Pg. 7, line 391)

Comment 4s: Line No 252: “quote:”.

Authors’ response: Thanks. This has been corrected in the paper. (Pg. 8, line 441)

Comment 4t: Line No 351: “Because”.

Authors’ response: Thanks. This has been corrected in the paper. (Pg. 10, line 541)

Comment 4u: Line No 370: “We”.

Authors’ response: Thanks. This has been corrected in the paper. (Pg. 10, line 561)

Comment 4v: Lines No 421-422: transfer the reference “[42]” at the end of the sentence.

Authors’ response: Thanks for the suggestion. We have moved the reference at the end of sentence.

Reviewer 5 Report

The authors investigated “Knowledge, beliefs and clinical practices of General Medical Practitioners towards oral cancer in Australia” with special regard to fast growing indian community and use of areca (betel) nut. Please revise title by including these informations. Overall, the manuscript is well written with adequate and novel references used. In M&M section, regarding the study design it could be interesting even to state why interview ranges of GP’s vary from 17 to 43 minutes. It should be emphasized that background of GP’s training seemed to be very diverging. Please discuss possibility of a qualified dental specialist accompanying screenings and checkups…Please describe whether oral investigations are performed by GP’s in any form and discuss whether it could make sense to be performed by them (needs training, or transfer to dental specialist).

Author Response

Thank you for providing us the opportunity to revise our manuscript for International Journal of Environmental Research and Public Health. Please find below the details of changes we have amended in the light of the reviewers’ comments. All changes in response to reviewer’s comments are indicated with track changes in the manuscript. It should be noted that some line numbers (after change in each page) are missing in the manuscript version with revisions (unfortunately couldn’t be corrected in this format due to track changes- the word document tries to be consistent with the line numbers even the ones which have been deleted). Therefore, the revisions have been addressed according to the page and line numbers in the current format (with track changes).

The following revisions have been made to the manuscript following reviewers’ comments:

REVIEWER 5:

Comment 1: The authors investigated “Knowledge, beliefs and clinical practices of General Medical Practitioners towards oral cancer in Australia” with special regard to fast growing Indian community and use of areca (betel) nut. Please revise title by including this information.

Authors’ response: Thanks for the recommendation. We have revised the title as follows - Perceptions and Practices of General Practitioners Towards Oral Cancer and Emerging Risk Factors among Indian Immigrants in Australia: A Qualitative Study.

Comment 2: Overall, the manuscript is well written with adequate and novel references used.

Authors’ response: Thank you for the positive feedback.

Comment 3: In M&M section, regarding the study design it could be interesting even to state why interview ranges of GP’s vary from 17 to 43 minutes.

Authors’ response: Thanks for the comment. Research studies involving GPs are very challenging due to their overly busy clinical routine and schedules. In our study, the interviews were conducted in semi-structured format at the times feasible for GPs and these time slots were usually between patient appointments or lunch times or while they were driving home or to work. This led to the varying lengths of interviews ranging from 17 minutes (shortest interview) to 43 minutes (longest interview). We now have stated this reason in the Materials and Method section. (Pg.4, line 241-243)

Comment 4: It should be emphasized that background of GP’s training seemed to be very diverging.

Authors’ response: Thanks for highlighting this. Since this was an exploratory qualitative study, we deliberately aimed to seek diversity in the sample (GP participants) while employing a combination of purposive and snowball sampling. The GP community in Australia is very diverse (Royal Australian College of General Practitioners, General Practice Health of The Nation. Available from: https://www.racgp.org.au/getmedia/c2c12dae-21ed-445f-8e50-530305b0520a/Health-of-the-Nation-2020-WEB.pdf.aspx) in terms of their training with a large proportion (51-80% depending on specialty training) of GPs having their basic qualification from overseas. Therefore, our purpose was to gain insights into the oral cancer related knowledge and practices of GPs’ who are from diverging training backgrounds. We have emphasized this in the results section of paper. (Pg. 5, line 315-318; Pg. 6, line 325-328). In addition, to help the reader better appreciate the diversity in training we have included an additional information in the sampling section outlining the various training options to become a GP in Australia. (Pg. 4, line 219-226)

Comment 5: Please discuss possibility of a qualified dental specialist accompanying screenings and check-ups.

Authors’ response: Thanks for the suggestion. The idea of inclusion of qualified dental specialist accompanying GP during oral cancer screening and check-ups is an excellent future recommendation from a policy perspective. We have inserted it as an implication in the discussion section of paper. (Pg.13, line 714-718). We have also added that although such shared models of care exist in other countries like the UK (NHS. Primary care commissioning.  [cited 2021 10th October]; Available from: https://www.england.nhs.uk/contact-us/privacy-notice/how-we-use-your-information/our-services/primary-care-commissioning/.) it will be more challenging in Australia due to the lack of a universal dental scheme.

Comment 6: Please describe whether oral investigations are performed by GPs in any form and discuss whether it could make sense to be performed by them (needs training, or transfer to dental specialist).

Authors’ response: Thanks for the comment. Since GPs are primary care providers and first point of contact in many developed countries, they are more likely to see patients with potential oral cancer lesions/disease. Hence it becomes more important for GPs to engage in duties of oral cancer prevention counselling, initial screening and routine oral/mouth check-ups (Ahern, J., et al., Building bridges with dentistry: NICE guideline supports collaborative practice between GPs and dentists. British Journal of General Practice, 2020. 70(698): p. 461-461. Available from: https://bjgp.org/content/bjgp/70/698/461.full.pdf). A collaborative practice thus could help in setting a standard referral pathway which can be employed to refer patients to a dental specialist in case of suspected oral cancer lesions for definitive diagnosis.

Guidelines for preventive activities in general practice by Royal Australian College of General Practitioners (https://www.racgp.org.au/download/Documents/Guidelines/Redbook9/17048-Red-Book-9th-Edition.pdf) has also emphasized oral cavity check-ups by GPs in clinics. In addition, although the general practice is overburdened, oral cancer-related training (Blaylock, P., R. Lish, and M. Smith, Oral health training for general practitioners and general practice teams. Education for Primary Care, 2020. 31(4): p. 240-243) can be provided to GPs through online resources (specifically about emerging risk factors) and short training modules to help them in identifying oral cancer in early stages. We now have refined the discussion section to include this information. (Pg. 13, line 724-726/ 732-734)

Round 2

Reviewer 2 Report

Thank you for your comprehensive revisions to the paper. It now has much stronger context that has improved the paper significantly. 

Reviewer 3 Report

All my concerns have been addressed